# Wind-driven device for cooling permafrost

Yinghong Qin[1,2], Tianyu Wang [2] & Weixin Yuan[2]

Preserving permafrost subgrade is a challenge due to global warming, but passive cooling techniques have limited success. Here, we present a novel wind-driven device that can cool permafrost subgrade by circulating coolant between the ambient air and the subgrade. It consists of a wind mill, a mechanical clutch with phase change material, and a fluid-circulation heat exchanger. The clutch engages and disengages through freezing and melting phase change material, while the device turns off when the outside air temperature exceeds a certain threshold, preventing heat from penetrating the subgrade. Two-year observations demonstrate that the device effectively cooled permafrost measuring 8.0 m in height and 1.5 m in radius by 0.6–1.0 °C, with an average power of 68.03 W. The device can be adapted for cooling embankments, airstrip bases, pipe foundations, and other structures. Further experimentation is required to evaluate its cooling capacity and long-term durability under various conditions.

Polar and high-altitude infrastructures pose significant challenges due to the increasing threat of permafrost warming, which compromises the subgrade integrity of buildings[1–3]. To ensure the bearing capacity of the subgrade, engineers have employed passive cooling methods to preserve the permafrost. One commonly employed passive cooling technology involves the regulation of convective heat loss from the ground and the utilization of cold ambient air to cool the permafrost subgrade, mitigating the effects of global warming. By enhancing gas convection during cold seasons to dissipate the internal heat of the subgrade, the bearing capacity can be effectively maintained. Notable techniques include the utilization of ventilation ducts[4–7], crushed rock embankments[8–10], and thermosyphons[1,11–14]. For instance, in the case of subgrades incorporating ventilation ducts, the thermal energy within the subgrade can be transferred to the cold air through the air ducts in cold seasons[5,15]. During warm seasons when wind strength is relatively weak, subgrades overlain by ventilation ducts experience a net cooling effect. In the context of a convective embankment with a layer of crushed-rock revetment, cold air permeates through the pores of the crushed rocks during cold seasons, while the hot air at the bottom of the layer rises, creating a circulation pattern. Air circulation drains internal heat from the subgrade through reverse convection cycles, but this cooling capacity may be compromised if voids between rocks are filled with sands or other solid matter[13]. Likewise, thermosyphons offer an effective means to cool subgrades. With the lower end inserted into the soil and the upper end exposed to the air, thermosyphons

operate by leveraging temperature differentials. When the ambient air temperature is colder than the working fluid within the lower end, the fluid undergoes boiling, and the resulting vapor rises to the upper end, effectively transporting heat from the soil (source) to the air (sink). The condensed vapor then falls back into the liquid pool, allowing the thermosyphon to operate continuously[16]. Conversely, when the air temperature exceeds that of the working fluid, condensation does not occur, resulting in a state of dormancy for the thermosyphon. In the thermosyphon system, heat from the surrounding soil is dissipated through the evaporative and condensing loop of ammonia. Due to the circulation of ammonia vapor, this process can still be classified as air convection cooling, adhering to the principles of heat exchange.

Traditional passive cooling techniques prove inadequate when applied to cool the subgrade of expressways in permafrost regions[17–21]. Both crushed rock layers and convection ducts exhibit low cooling efficiencies that fail to adequately preserve the underlying permafrost in the face of global warming. Furthermore, temperature variations exist between the center and edges of the crushed rock layer. While thermosyphons demonstrate relatively robust cooling capabilities, their effectiveness relies on vertical or slightly slanted insertion angles. Significant tilting compromises the cooling capacity of thermosyphons[11]. To enhance the cooling intensity of permafrost subgrades, researchers have explored the cooling effects of combining different cooling technologies[19,22–26]. Although the combined use of passive cooling techniques partially addresses the limitations of

[1]School of Civil Engineering and Architecture, Guangxi Minzu University, 188 University Road, 530006 Nanning, China. [2]College of Civil Engineering and Architecture, Guangxi University, 100 University Road, 530004 Nanning, China. ✉e-mail: tywang@st.gxu.edu.cn

individual methods, it does not qualitatively improve the strength of permafrost cooling to preserve permafrost under expressway. The concept of cooling permafrost subgrades remains confined to air convective cooling approaches. Since the 1970s, efforts have been made in North America to reduce the temperature of frozen soil beneath subgrades by increasing road surface reflectivity. However, the use of high reflectivity coatings and slippery pavements has been hindered by durability issues, preventing the widespread adoption of high reflectivity pavements[27–29]. Recent advancements in synthetic asphalt-based and resin-based materials have paved the way for high albedo pavements[8,30]. Revisiting the use of high-albedo pavements to mitigate permafrost degradation beneath subgrades has emerged as a topic of interest[3]. However, it has been observed that ultraviolet radiation ages the reflective pavement surface within 1–2 years. Consequently, enhancing passive cooling capacity to preserve permafrost subgrades of expressways has become a recent research focus[19,20,26,31,32]. In an effort to dissipate heat from roadbeds during summer, Liu et al.[33] proposed a method of cooling permafrost roadbeds using a steam compression refrigeration system. This system, akin to manual freezing or forced cooling using wind or solar energy, involves the use of electricity to transfer heat from the permafrost layer to the external environment. However, the durability of the device remains uncertain as the forced convection relies on electrical power.

The current approach to cooling permafrost subgrades primarily relies on air convection cooling, which exhibits limitations due to the low specific heat of gases and consequently lower efficiency in heat transfer through convection. The potential for increasing the strength of cooling frozen soil within this framework is constrained. It is imperative to explore alternative cooling techniques that surpass the constraints of air convective cooling and pave the way for high-efficiency cooling technologies. Here, we introduce a promising approach utilizing liquid convection for controlled heat exchange between the permafrost stratum and ambient cold air, ultimately resulting in the cooling of the permafrost. By automatically shutting off the pump when the air temperature reaches a specified threshold, the cooling capacity can be significantly enhanced. This approach elevates the heat carrier from air to liquid, thereby enabling a transition from natural convection to forced convection. Additionally, the forced convection of the liquid allows for targeted circulation, enabling the cooling method to effectively cool wide subgrades, pile foundations, and other relevant subgrade areas.

## Results

In this section, the cooling capacity of the device is demonstrated through the presentation of the temperatures of the coolant, of the device's outer skin, and of the surrounding soils. While the temperatures of the soils are recorded on an hourly basis, the mean daily temperature is presented unless otherwise specified.

### The wind-driven cooling device operates effectively

The wind-driven cooling system has been installed at the Beiluhe Permafrost Station in the Qinghai-Tibet Plateau for over 2 years. The local weather conditions vary between winter and summer, with mean daily air temperatures dropping below −2 °C for several months during the winter, and occasionally falling below this temperature at night-time during the summer (Supplementary Information Figs. S1 and S2). If there is a significant temperature difference between the coolant in the inner tube and that in the outer annulus, it indicates that the pump is effectively circulating the coolant. This means that thermal energy is being transferred from the soil to the outer tube, indicating that the clutch is engaged. However, if the temperature difference is minimal, the clutch remains disengaged.

The findings from the measurements reveal that the duration of clutch engagement exhibits daily, seasonal, and annual variations, with

the highest engagement observed during the months of January and February. The daily fluctuations in clutch engagement duration provide strong evidence of the successful operation of the wind-driven cooling system. The temperature profiles of the inner tube and outer annulus depicted in Fig. 1a–c indicate that, on a typical day in autumn, winter, and spring, the clutch is engaged, and the coolant circulates through the tube. In contrast, on a typical day in summer, the temperature profiles of the inner tube and outer annulus almost overlap (Fig. 1d), indicating that the clutch is disengaged, and the device is in hibernation. The consistency of these temperature profile patterns serves as a testament to the device's successful operation in the field, in line with expectations.

The temperature difference along the double tube is derived with respect to the depth. It is evident that the temperature along the inner tube exhibits negligible variation, indicating a minimal loss of thermal energy along the tube. This desirable characteristic of minimal heat flux is advantageous, especially considering that the inner tube is constructed from PVC material, which effectively minimizes energy loss during the device's operation. At the outer annulus, the temperature of the coolant from the bottom of the device increases progressively until to the 1.0 m depth. The power of the device ($q$) is the product of the heat capacity ($c$), density ($\rho$), instantaneous discharge rate ($v$), and temperature change ($\Delta T$) of the flowing coolant along the outer annulus, that is $q = c\rho v \Delta T$. The heat capacity and density of the coolant can be found in Table 1, while the instantaneous discharge rate ($v$) can be found in Fig. S5 in Supplementary Information. It is found that peak instantaneous power outputs reach an impressive 300 W (Fig. 1j). An annualized assessment suggests the device's potential to consistently deliver an average power of 68.03 W. It is imperative to acknowledge that such power metrics are susceptible to variations influenced by the phase change material's (PCM) freezing temperature within the clutch, the pump's operational parameters, and prevailing ambient air temperatures.

The operational characteristics of the device exhibit daily and seasonal variations. By utilizing the temperature profile separation between the inner tube and outer annulus, we have successfully determined the duration of coolant circulation for each date between September 1, 2020 and September 1, 2022. Our analysis reveals that the device operates from mid-October to mid-April, with sporadic operational hours observed outside of this time frame. During the middle of winter, when the local air temperature remains below −10 °C, the device's full-day operation is curtailed to limited durations due to intermittent winds. From early July to early October of the subsequent year, the device remains fully disengaged. In the course of this experiment, the annual operational duration for the year 2021 was determined to be 105.15 days, which corresponds to approximately 0.288 of the entire year. It is worth noting that the device enters a hibernation state when the temperature of the phase change liquid within the clutch exceeds its freezing point −4 °C. In an examination of phase change materials (PCMs), elevating the freezing point corresponds to a prolonged operational duration. Conversely, a lower freezing point truncates the device's operational window. However, for instance, increasing the freezing temperatures somewhere between −3.5 °C and −1 °C, while extending operational time, may inadvertently impart heat to regions that have undergone cooling to temperatures below −3.5 °C.

### Temperature serials of the surrounding soil validate the device's cooling efficiency

The remarkable cooling capabilities of the device are vividly demonstrated by the temperature series recorded at various depths and distances. Fig. 2a portrays the temperature fluctuations within the soil at a depth of 2.0 m, presenting data collected at distinct radii from the centerline of the device. The radii examined include 0.04 m, 0.60 m, 1.50 m, 3.0 m, and 5.7 m. Notably, during the period from

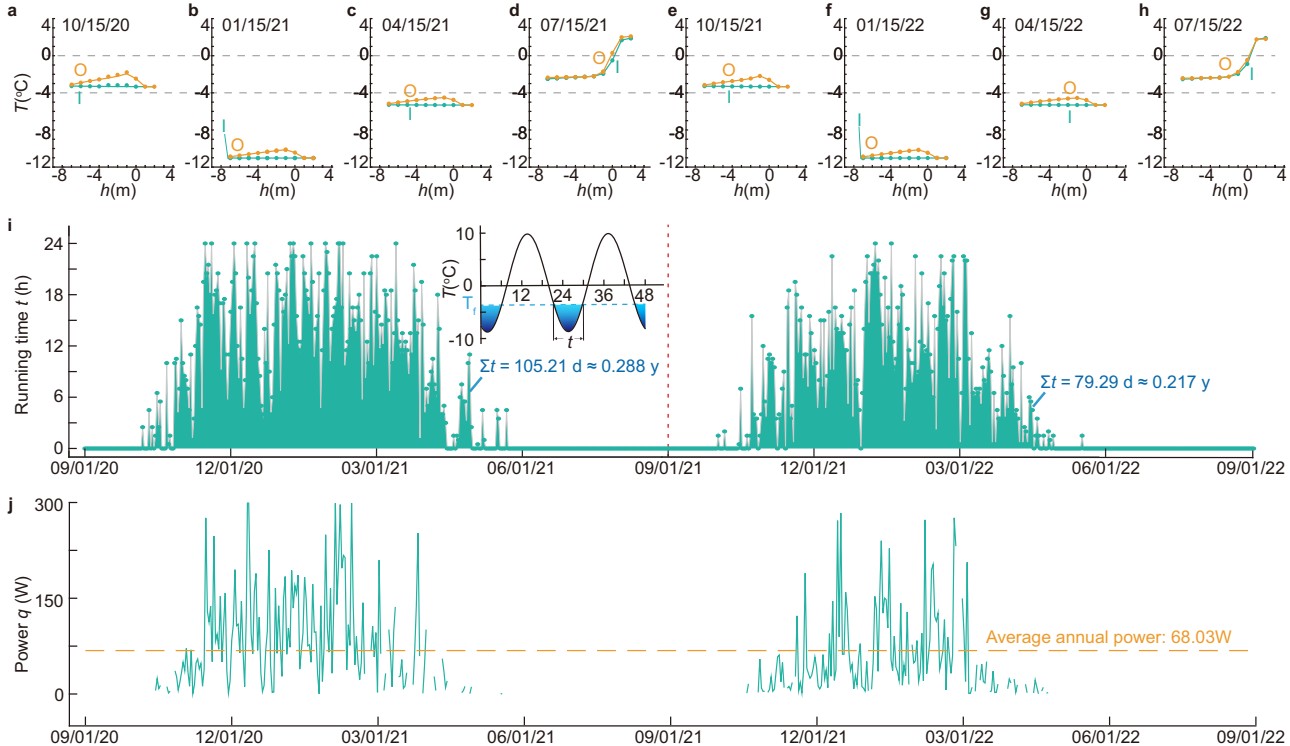

**Fig. 1 | Temporal engagement patterns of the device's clutch. a–h** Temperature profiles of the inner tube and outer annulus across representative dates over two operational years. **i** Operational duration spanning the two-year period. **j** Device power metrics throughout the biennial operation. Note: I inner tube, O outer tube. Source data are provided as a Source data file.

mid-May to early October, the device's skin temperature remains relatively stable at a depth of 2.0 m, primarily owing to the device's hibernation state throughout most of this duration. It is essential to emphasize that the temperatures observed in 2022 surpassed those of the preceding year, largely due to local weather conditions exerting their influence. Of particular interest is the device's skin temperature at a radius of 0.04 m, which experiences considerable variability during winter due to the daily fluctuations in the local mean air temperature. Moreover, this temperature series exhibits a discernible seasonal pattern, influenced directly by the temperature of the circulating coolant. In contrast, at a depth of 2.0 m, the temperatures of the surrounding soils at radii of 3.0 and 5.7 m converge closely, suggesting minimal impact from the device on the soil located at these greater distances. Comparatively, the temperature variations observed at radii of 0.6 and 1.5 m surpass those at 3.0 and 5.7 m, albeit to a lesser extent than the fluctuations seen at radii of 0.04 and 0.6 m. Consequently, the device effectively cools the surrounding soil up to a radius of 1.5 m. This finding underscores the impressive reach of the device's cooling influence in the immediate vicinity.

Further validation of the device's ability to cool the surrounding soil within a radius of up to 1.5 m can be derived from the temperature series recorded at depths of 4.0, 6.0, and 8.0 m, as depicted in Fig. 2b–d. Remarkably, at these depths, the soil temperature exhibits minimal susceptibility to the seasonal fluctuations occurring at the

ground surface. Consequently, the temperatures of soils situated at radii of 3.0 and 5.7 m from the device remain consistently stable throughout the measurement period, as exemplified in Fig. 2b–d. However, the soils located at radii of 0.6 and 1.5 m experience a notable cooling effect over time, as evidenced by the corresponding temperature profiles displayed in Fig. 2b–d. This observation further underscores the robust cooling impact of the device. It is worth noting that, in the first year, the soil at a radius of 1.5 m surrounding the device undergoes a substantial cooling process. While this soil continues to experience cooling in the second year, the additional decrease in temperature becomes relatively minor. This cooling pattern implies that the device is capable of effectively and rapidly cooling the surrounding soils within a radius of 1.5 m within a one-year timeframe.

The device's skin temperature at a depth of 8.0 m, as depicted in Fig. 2d, displays a slightly diminished level of variability compared to the skin temperatures observed at the other two depths, illustrated in Fig. 2b, c. This reduction in temperature fluctuation can likely be attributed to the influence of both the surrounding soils and the coolant present in the outer annulus. Notably, the coolant temperature reaches its lowest point at the bottom of the outer annulus, while it attains its highest value at the top. In contrast, the soil at a depth of 8.0 m remains relatively warmer when compared to the soils situated at depths of 4.0 and 6.0 m. Consequently, the device's skin temperature at depths of 4.0 and 6.0 m exhibits a notable degree of variation, likely due to the interplay between the fluctuating coolant temperature and the differing thermal characteristics of the surrounding soils. This disparity in temperature fluctuations further underscores the complex dynamics at play within the system.

A comprehensive assessment of the device's cooling capabilities can be gleaned from the temperature profiles obtained from the surrounding soils. Fig. 2e–l present the temperature profiles of soils located at distances of 0.04, 0.6, 1.5, 3.0, and 5.7 m on selected dates during the experimental measurement period. Notably, in autumn, the

## Table 1 | Thermal properties and geometry of fluid-circulation heat exchanger

| Components | Thickness (mm) | Diameter (mm) | Density $\rho$ (kg/m³) | $C_p$ (J/g/K) | $k$ W/(m·K) | $T_l$ (°C) |
|---|---|---|---|---|---|---|
| Inner tube | 3 | 33 | 1380 | 900 | 0.14 | – |
| Outer tube | 4 | 88 | 7850 | 470 | 52.34 | – |
| Coolant | – | – | 1058 | 3121 | 0.34 | −45 |

Note: $T_l$ = phase change temperature, $C_p$ = specific heat capacity.

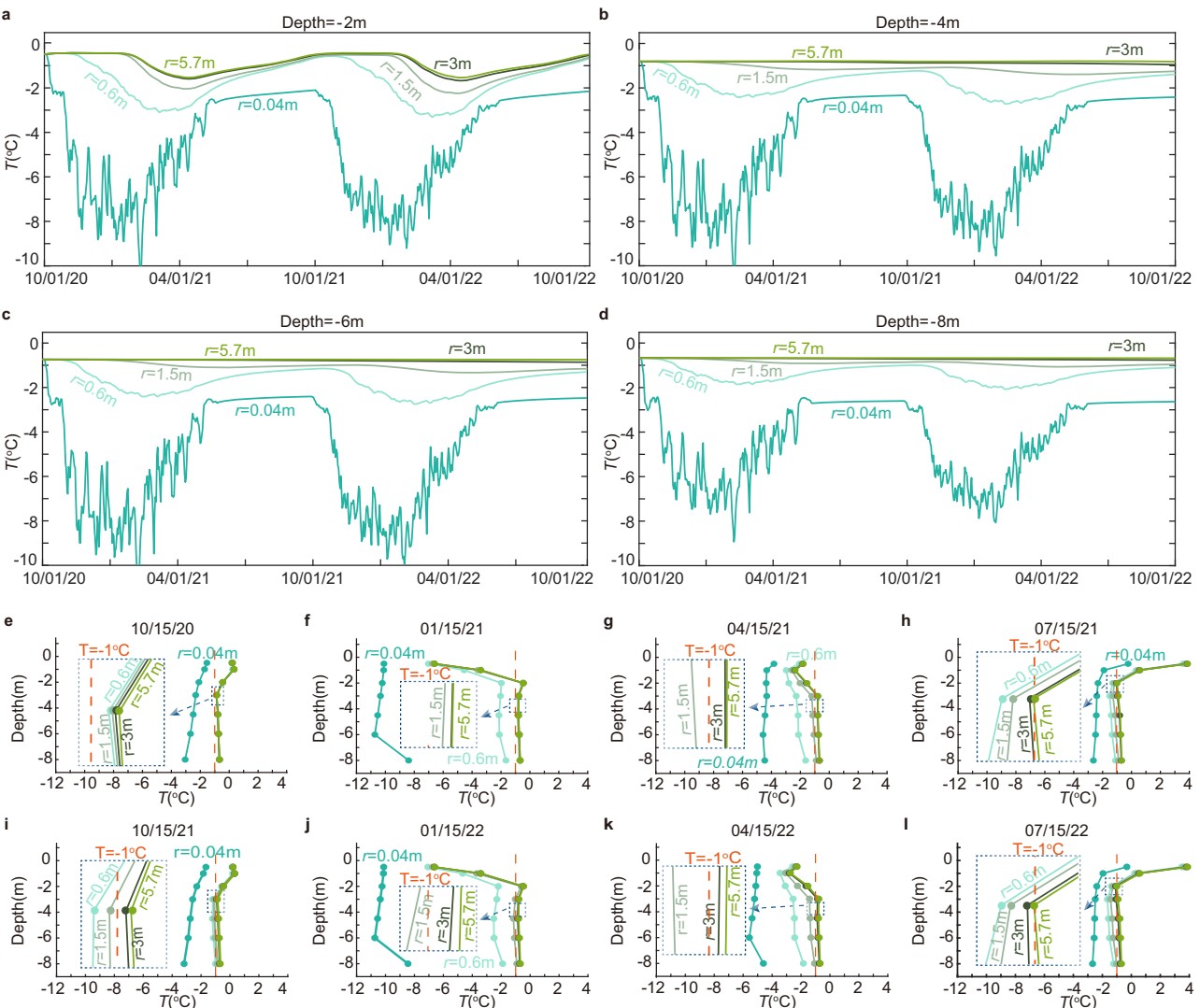

**Fig. 2 | Soil temperature series and profiles corroborate the device's cooling effect. a–d** Soil temperature series across varying depths and radii proximate to the device. **e–l** Temperature data captured on selected dates during the experimental observation. Source data are provided as a Source data file.

temperature profile of the soil adjacent to the device descends below 2 °C, while the remaining temperature profiles tend to converge, indicating that the device has been in hibernation for several months (Fig. 2e, i). During the winter season, the temperature profiles exhibit significant divergence, primarily due to the prolonged circulation of the coolant within the double tube system (Fig. 2f, j). Following a year of operation, the temperature profiles at distances of 0.04, 0.6, and 1.5 m display distinct separations (Fig. 2i–l). In contrast, the temperatures at distances of 3.0 and 5.7 m align closely, further validating the device's effective cooling influence within a radius of up to 1.5 m, while the cooling effect has minimal impact beyond this range. This observation provides compelling evidence regarding the localized nature of the device's cooling capacity and reinforces the notion that its influence is primarily confined to the immediate surroundings, up to a radius of 1.5 m.

## The cooling cylinder around the device enlarges over time

The temperature contour surrounding the device, depicted in Fig. 3, offers valuable insights into its cooling dynamics on selected dates, utilizing linear interpolation. Notably, on October 15, 2020, approximately one and a half months after installation, the soil column within a

radius of 0.5 m around the device experienced a cooling effect, reducing the temperature from approximately −0.7 °C to −1.0 °C (Fig. 3a). By January 15, 2021, this cooled column had expanded outward, reaching an extent of approximately 1.2 m, as indicated by the −1.0 °C iso-temperature contour (Fig. 3b). Subsequently, on April 15, 2021, further expansion was observed, encompassing the soil column with a radius of 1.5 m, as illustrated in Fig. 3c. By July 15, 2021, a slight reduction in the cooling rate became apparent, evidenced by the retraction of the −2.0 °C iso-temperature contour closer to the device compared to its position on April 15, 2021. However, the −1.0 °C iso-temperature contour had extended beyond the column with a radius of 1.5 m, providing further confirmation of the device's effective cooling capacity within a soil column extending up to 1.5 m around the device. These findings validate the device's ability to efficiently cool the soil column within a radius of 1.5 m surrounding the device, as demonstrated by the spatial extent of the temperature contours on the selected dates.

In the subsequent year, the temperature contour analysis reveals the device's continued cooling impact on the surrounding soils (Fig. 3e–h). A comparison between the same dates in 2021 and 2022 demonstrates that the −1.0 °C iso-temperature contour has extended

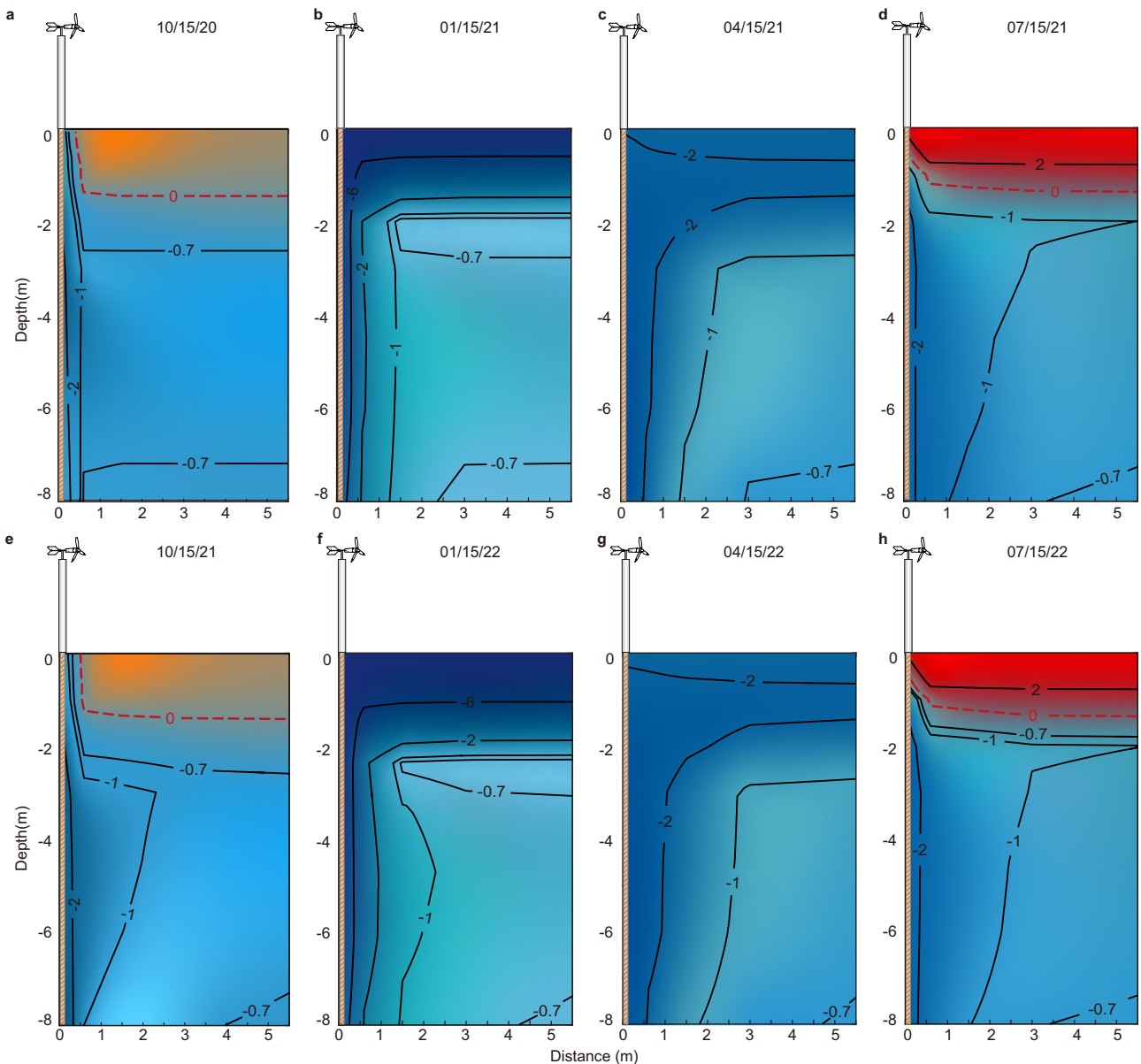

**Fig. 3 | Temperature contours of the soil column adjacent to the wind-driven cooling device on key dates. a** On 15 Oct. 2020, **b** on 15 Jan. 2021, **c** on 15 Apr. 2021, **d** on 15 Jul. 2021, **e** on 15 Oct. 2021, **f** on 15 Jan. 2022, **g** on 15 Apr. 2022, **h** on 15 Jul. 2022. Source data are provided as a Source data file.

further away from the device. Additionally, the area encompassed by the −0.7 °C iso-temperature contour has noticeably diminished. These variations in the extent of these two iso-temperatures provide compelling evidence of the device's ongoing ability to cool the surrounding soils. Nevertheless, the magnitude of this difference diminishes with the passage of time. An intriguing finding arises from the observation that, while the ground temperature from 2 to 8 m has been effectively cooled, the place of the permafrost table remains unchanged. This observation bears significance as an upward movement of the permafrost table can lead to frozen heave of the shallow ground. Therefore, the preservation of the permafrost table within this context may be beneficial, ensuring the stability of the shallow ground and mitigating potential concerns associated with frozen heave.

## Discussion

The wind-driven cooling device exhibits parallels with apparatuses delineated in prior research (Table 2). Specifically, Zhu et al. introduced an instrument that employs a windmill to channel cold air through pre-established conduits, thereby cooling embankments in permafrost domains[34]. Although the utilization of wind and cold air as cooling mechanisms for permafrost strata bears resemblance, the mechanism delineated by Zhu et al. diverges fundamentally from our presented wind-driven cooling device. Firstly, our model facilitates permafrost cooling via a sealed circulation loop wherein the heat carrier functions as a liquid coolant. In contrast, rendition of Zhu et al. employs an open system, with air acting as the heat carrier. Secondly, our device's operationality hinges on the liquid–solid phase transition of a meticulously formulated alcoholic solution, whereas the system posited by Zhu et al. is potentially electrically actuated, although such specifics remain unspecified in their publication.

The device in this study exhibits parallels with thermosyphons, though they utilize divergent cooling pathways (Table 2). Both systems deploy cylindrical tubes embedded into the soil, serving as conduits for heat transfer. Distinctions between them emerge in aspects such as the heat carrier, installation orientation, convection modality, and monitoring methodologies (Table 2). Crucially, the wind-driven cooling device presents three salient benefits when contrasted with thermosyphons. Firstly, the device demonstrates

**Table 2 | Comparative analysis of device parameters with previously published devices**

| Cold tools | Heat carrier | Install angle | Heat transfer type | Detection | Switch |
|---|---|---|---|---|---|
| The device | Liquid coolant | Any angle | Force convection | Hall sensor | Automatically |
| Thermosyphon | Vapor | 0–90° | Convection, evaporation, and condensing | Difficult | Automatically |
| Zhu et al.'s device | Air | – | Force convection | – | – |

Note: (1) The thermosyphon's cooling capacity is significantly diminished at reduced installation angles. (2) Thermosyphon heat transfer is more complex than natural air convection due to the accompanying processes of evaporation and condensation. (3) "–" = do not mention.

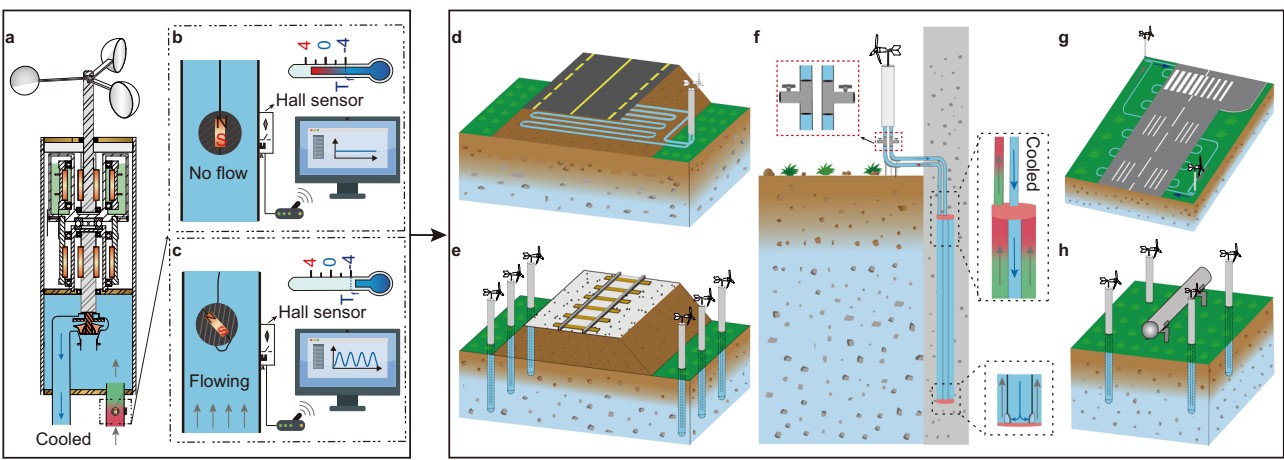

**Fig. 4 | Illustrative schematics highlighting diverse applications of the device. a–c** Employment of Hall-effect sensors for remote detection of coolant circulation within the tube. Device applications encompass **d, e** roadway embankments, **f** pile foundations, **g** airstrip bases, and **h** pipeline infrastructures.

higher heat transfer efficiency owing to its utilization of forced fluid circulation, which proves more efficient compared to the air convection employed by thermosyphons (Table 2). As illustrated in Fig. 1j, the device registers an annual cooling power of 68.03 W, while the experiment conducted by Zhang et al. show that the instantaneous power of an active thermosyphon was recorded to be less than 25 W (see Fig. S5 in Supplementary Information)[11]. Factoring in dormant and active periods throughout the year, the annual output of an operational thermosyphon does not exceed 25 W. Secondly, real-time monitoring of the device's operational status is made possible through the integration of a magnetic floater and a hall sensor (as depicted in Fig. 4a–c). The magnetic floater possesses a density that is slightly lower than that of the coolant, enabling its displacement from the original position as the coolant circulates within the tube. Subsequently, the outer hall sensor detects changes in the magnetic field, transmitting a signal to a remote monitoring system in the laboratory. This innovative approach allows for constant monitoring of the device's functionality. For instance, when the local temperature drops below −10 °C and the local wind speed ranges from 3–6 m/s, the coolant should circulate through the tube. Failure to receive any signal at the monitoring end indicates a malfunction in the device, necessitating inspection and repair. Conversely, in the case of a thermosyphon, it becomes challenging to ascertain the flow of vapor and condensation when the external air temperature is colder than that of the subterranean soils. Lastly, the wind-driven cooling device exhibits enhanced adaptability in deployment due to the ability to separate the fluid-circulation heat exchanger into two parts, connected by flexible tubes. This configuration facilitates the convenient transportation of coolant to desired locations. Moreover, the separation of the heat exchanger into harvesting and depositing ends enhances the device's versatility (as illustrated in Fig. 4a). In contrast, while a thermosyphon may be utilized at oblique angles relative to the vertical axis, achieving a

horizontal or inverted position for the vaporizing end (heat-absorbing end) becomes arduous.

Notably, the current device configuration presents certain limitations that warrant further investigation. The device facilitates mechanical load transfer from the windmill through a magnetic non-contact mechanism. While this approach effectively minimizes frictional losses, the intricacy of the mechanical transfer system poses questions about the long-term durability of components, such as the bearings, necessitating further field evaluations. Presently, the system incorporates two magnetic couplings, adding to its complexity. Future iterations aim to simplify this by employing a singular magnetic coupling. Given the device's anticipated field life expectancy of over 50 years, assessing the long-term durability of the pump remains crucial. Intuitively, the pump's impeller may be susceptible to fatigue, especially when operating at elevated rotational speeds over extended periods. Given that the probability of fatigue failure diminishes exponentially with reduced rotational speed, a forward-looking approach might entail the design of a pump optimized for high discharge rates at lower rotational velocities.

In a departure from its conventional use as an alternative to thermosyphons, the wind-driven device offers a wider range of applications. While it can indeed be positioned at the slope toe of an embankment to facilitate localized soil cooling—mirroring thermosyphons—the device is versatile enough for thermal management of diverse infrastructures including roadway embankments, piles, airstrips, and oil pipelines (refer to Fig. 4d–h). Notably, the fluid-circulation heat exchanger need not be restricted to a double-tube configuration. It can be coupled with a U-type tube, where one extremity interfaces with the pump's inlet and the opposing end with the outlet. For instance, positioning a U-type cooling tube beneath a roadway embankment prior to its filling can lower the embankment's temperature. By separating the fluid-circulation heat exchanger, the end responsible for draining heat of permafrost is anchored centrally to the concrete cage of the drilling pile before pouring concrete to the

drilling hole, with the inlet and outlet connectors protruding from the borehole (as depicted in Fig. 4f). These connectors are then linked to the cold end, followed by a mechanical clutch and the windmill. After the poured fresh concrete hardens, the tube will concentric with the concrete pile. In the process of cement hydration, there is an observed increase in temperature within the surrounding permafrost. Consequently, an intriguing avenue for exploration might be the circulation of a low-temperature coolant within the concrete pile through designated inlet to outlet points. This could potentially expedite the re-freezing of the adjacent soils. As demonstrated in Fig. 3, the device effectively cools a column with a radius of 1.5 m, resulting in a temperature drop of 0.6–1.0 °C after two years of installation. Notably, this column size is three times larger than that of a typical pile in geotechnical construction, enabling the device to maintain the frozen state of the soil surrounding the drilling pile. Contrarily, conventional cooling approaches for drilling piles necessitate at least four thermosyphons encircling the pile because this quartet configuration ensures a symmetrical thermal regime and effectively maintains the adjacent soil in a frozen state. Furthermore, by dividing the fluid-circulation heat exchanger, the heat-absorbing end can be employed as a sheet tube beneath embankments, airstrip bases, and similar foundations to preserve the underlying permafrost (refer to Fig. 4d–h).

This study presents on a two-year experimental investigation of the cooling performance of a wind-driven cooling device in a permafrost region. It is found that the device effectively cools the surrounding soils with a radius of 1.5 m after one year operation. While the cooled column in the second year extends further outward, the

extending rate diminishes. Despite the promising cooling performance, further research is necessary to comprehensively explore the device's potential. Further study is needed to assess the long-term cooling performance, as well as the effects of variations in parameters such as PCM, length, pump, radius, and others. Additionally, it is essential to subject the device to testing in various permafrost regions characterized by differing wind patterns and temperature fluctuations to assess the uniformity of its cooling effects. Further experimentation is also needed to evaluate the effect of device's length on cooling performance and to conduct numerical simulations of the entire device. The authors acknowledge that the optimal installation and customization of the device for site-specific conditions remain unknown. Moreover, the device must be tested over 30–50 years in harsh permafrost conditions. Despite these challenges, the authors believe that the wind-driven cooling device has significant potential and is adaptable to various conditions.

## Methods

### Introduction of a wind-driven device for cooling permafrost

Here we introduce a novel device aimed at preserving the permafrost subgrade (Fig. 5). The device hibernates during warm weather (Fig. 5b–e); in cold weather, wind is harvested to circulate coolant in the device, which drains heat from the permafrost stratum to the outer cold atmosphere (Fig. 5f–i). The device consists of three main components: a wind mill, a mechanical clutch, and a fluid-circulation heat exchanger. The wind mill is connected to the mechanical clutch, which is a critical component made of solid-liquid phase change material

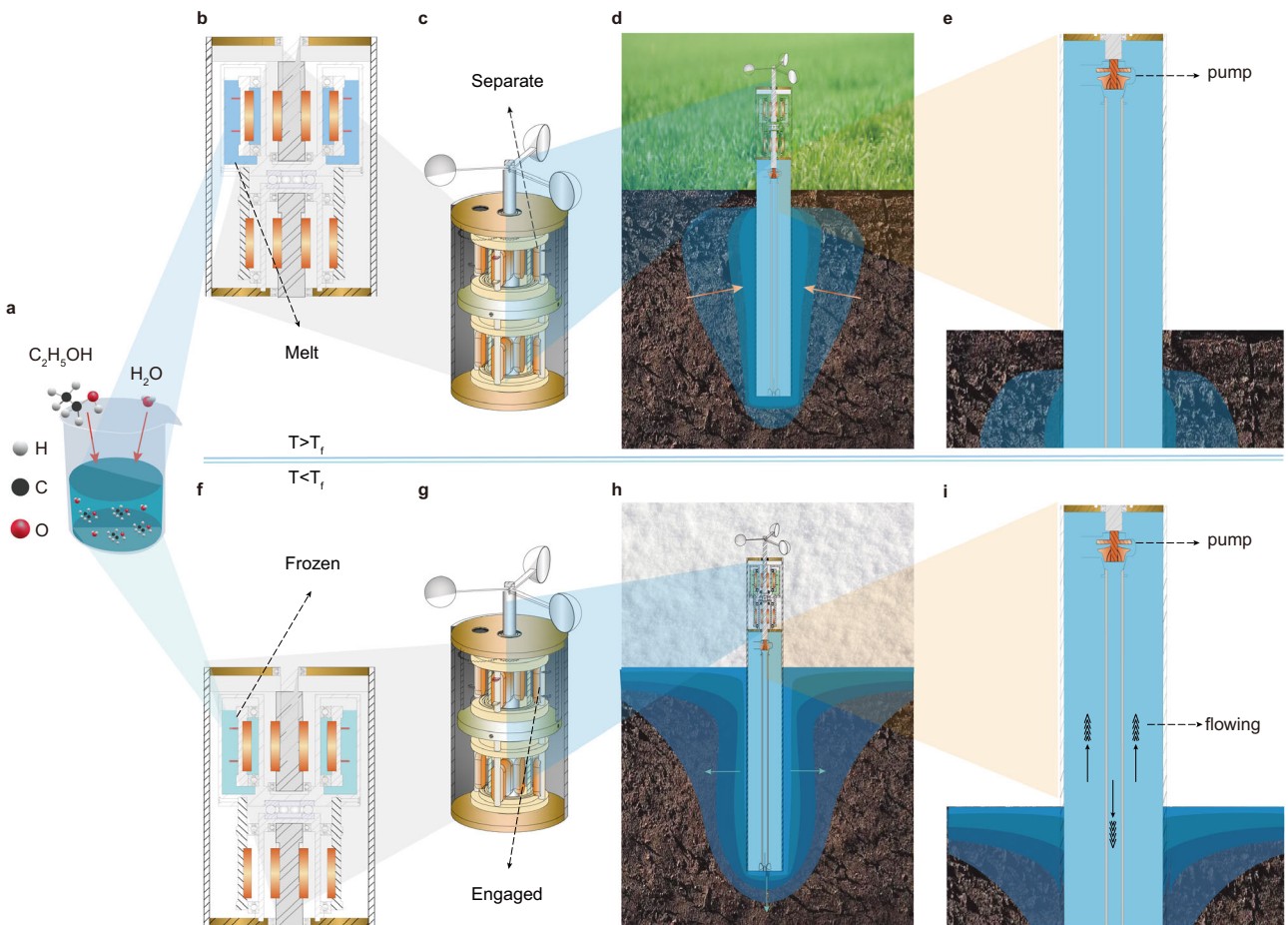

**Fig. 5 | A wind-driven cooling device for cooling permafrost stratum. a** An alcohol solution is designed to drop the freezing temperature of the solution some degrees below zero; **b–e** at warm weather, the device hibernates. **f–i** Wind is harvested to drive coolant circulating between permafrost ground and outer atmosphere. Note: $T_f$ = the freezing point.

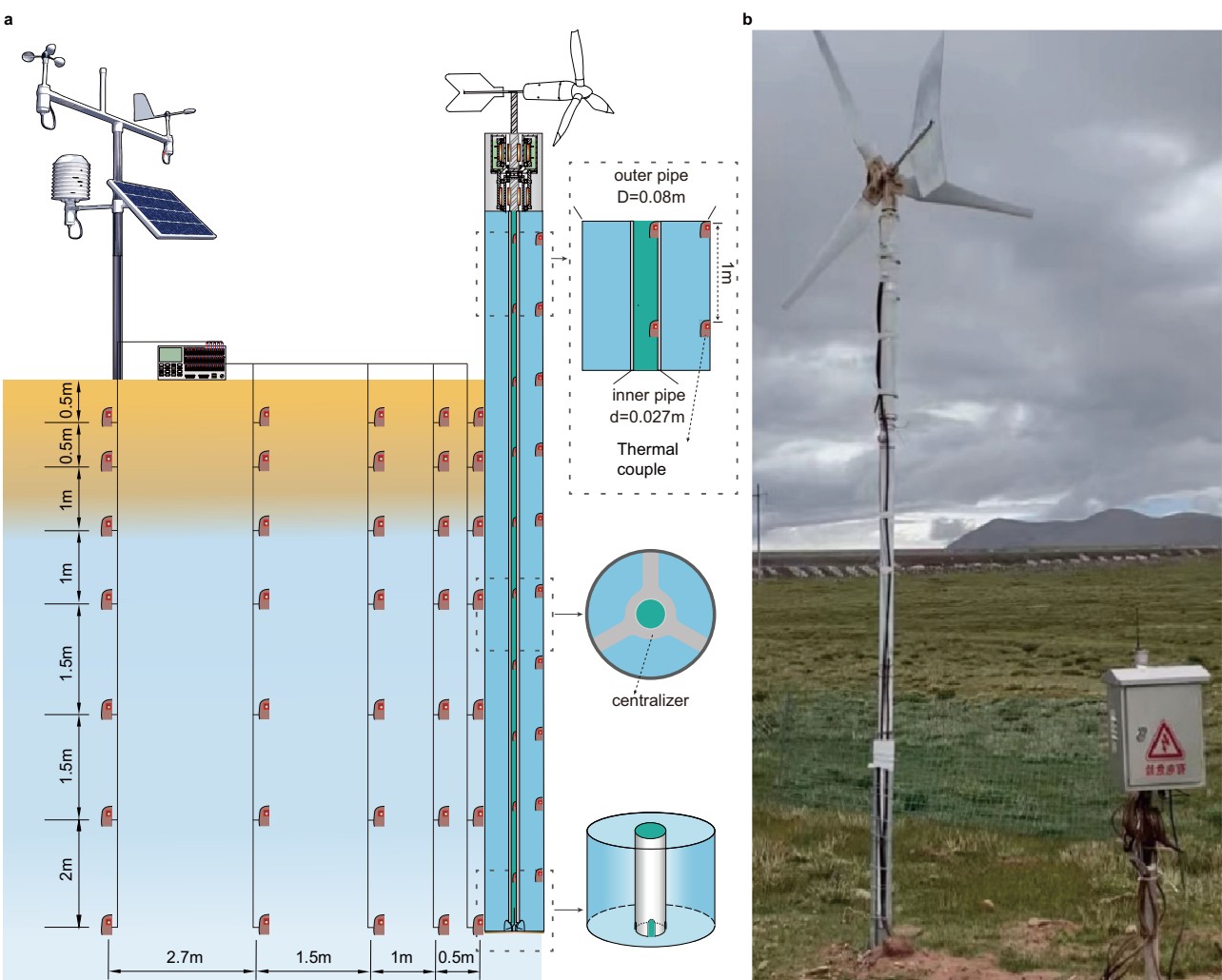

**Fig. 6 | Experimental configuration. a** Schematic representation of the experimental setup detailing thermometer placement. **b** Implementation of a wind-driven cooling device at the Beiluhe Permafrost Station, proximate to the Qinghai Tibet Railway.

(PCM). The fluid-circulation heat exchanger has a double-tube structure consisting of a large annulus enclosed concentrically with a small inner tube. The exchanger is filled with coolant and equipped with a pump. The inner tube features openings at its bottom that allows the coolant to circulate upward from the annulus between the outer and inner tubes, then return to the top of the inner tube (magnified in Fig. 6a). The end of the tubular exchanger equipped with a pump is connected to the clutch and exposed to the outer air a few meters, while the other end is buried underground (as shown in Fig. 6a).

The device described in the previous paragraph operates based on the temperature of the air surrounding it. In cold weather, when the air temperature falls below the freezing temperature of the Phase Change Material (PCM) used in the mechanical clutch, the PCM solidifies, engaging the clutch and allowing wind power to be harnessed and transmitted to the pump at the top of the fluid-circulation heat exchanger. The fluid-circulation heat exchanger was designed to be fully enclosed to prevent fluid evaporation or leakage during operation. To prevent the circulating fluid from evaporating or leaking, the windmill's rotating shaft is connected to the pump's rotating shaft using a non-contact magnetic gear. Both shafts are equipped with several magnetic pairs that are aligned concentrically. These magnetic pairs do not touch directly; instead, they have a gap between them. The load transfer between the shafts is achieved through the magnetic attraction and repulsion forces. This design ensures that the pump and the circulating fluid remain sealed within a closed space. As the coolant

circulates through the heat exchanger, it absorbs heat from the surrounding soils and then drains this heat to the cold ambient air through forced convection within the outer tube of the tubular exchanger. However, in warm weather when the air temperature rises above the freezing temperature of the PCM, the PCM melts, disengaging the clutch and halting the fluid circulation. This suspension of fluid circulation occurs even in windy weather, effectively ceasing the convection heat transfer. This device serves as a practical solution for efficiently harvesting and storing low-temperature thermal energy, providing a more effective alternative to traditional air-convective cooling techniques. While the fluid-circulation heat exchanger can be designed at other curved shape (not necessary double-tube), the device can be adapted to cool wide subgrades, pile foundations, and other similar structures in an environmentally-friendly manner.

## Experimental setup

In our study, we constructed a device designed for cooling applications. The device consisted of a windmill with three blades, whose diameter was 0.8 m. A wind vane was integrated into the windmill to ensure that the blades faced the wind, regardless of its direction (Fig. 6). The windmill started to operate when the wind speed exceeded 1.8 m/s and under no load, could reach a rotational speed of 1200 r/min at 6 m/s wind. The mechanical clutch of the device utilized a 2% ethanol solution in deionized water as the solid-liquid phase change material (PCM) to lower the phase change temperature to −4 °C. The fluid-circulation heat

exchanger was designed with an inner PVC tube, which had low thermal conductivity to minimize the heat gain during the downward flow of the coolant. The outer tube was constructed from Q235 steel to improve heat transfer between the upward flow of the coolant and the surrounding soils. Further details regarding the thermal properties of the fluid-circulation heat exchanger can be found in Table 1. The tubular geometry of the exchanger was optimized such that the cross-sectional area of the inner tube was several times smaller than that of the outer annulus. This geometry allowed for a faster coolant flow in the inner tube, while a slower flow in the annulus, allowing more time for the heat of the surrounding soils transferring to the coolant. The pump, when linked to the wind mill, had a discharge rate of 200 mL/s at a wind speed of 10 m/s (more details about the discharge rate can be found in Supplementary Information).

The wind-driven cooling device was installed at the Beiluhe Permafrost Station on the Qinghai-Tibet Plateau in China (34.8° N, 92.9° E). The site is covered by grassland, featuring distinct geological strata, including a superficial layer of loose silt clay (0–1.0 m), followed by silt clay (1.0–2.0 m), icy clay (2.0–8.0 m), and, finally, a weathered mudstone subgrade. The local permafrost table is located at an approximate depth of 2.0 meters and the area is classified as a warm, ice-rich permafrost region with a mean annual ground temperature of −1.0 °C to −0.5 °C at depths of 10–15 meters. To monitor the temperature inside and outside the device, thermometers were installed. Inside the fluid-circulation heat exchanger, thermometers were placed at 1.0-m intervals to log the fluid temperatures of the inner pipe and the outer annulus. With a height of 10.0 m, a total of twenty thermometers were placed in the tubes. Outside the device, thermometers were mounted on the outer skin of the device at intervals of 0.5–2.0 m (Fig. 6). To minimize the impact of solar radiation on the freezing of the PCM within the device, a low-reflective coating was uniformly applied to its surface.

Prior to the installation of the wind-driven cooling device, a borehole with a diameter of 12 cm was drilled 8.1 m deep from the natural ground surface. The device was carefully centered within the borehole and secured in place by overhanging it so that its lower end was buried 8.0 m underground. The annulus between the borehole and the device was then filled with dry fine sand and compacted in layers. To measure the temperatures at different depths, four additional boreholes were drilled circumferentially around the primary borehole at distances of 0.6, 1.5, 3.0, and 5.7 m. PVC tubes equipped with thermometers were inserted into these boreholes, which were then filled with dry fine sand. The thermometers had a precision of 0.1 °C. The installation was completed on August 25, 2020 and all temperatures were logged at 60-min intervals using a Campbell CR3000 data logger with an AM16/32B 32-Channel Relay Multiplexer. The data logger began recording on September 1, 2020 and the collected data was remotely transferred. The measured temperatures from September 1, 2020 to September 1, 2022 are presented in this study.

## Data availability
Source data are provided with this paper.

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

## Author contributions

Y.Q. conceived and supervised the project, and reviewed and edited the manuscript. T.W. developed and applied the methodology, performed the formal analysis and investigation, and wrote the original draft of the manuscript. W.Y. conducted the formal analysis, curated the data, and reviewed and edited the manuscript. All authors contributed significantly to the work.

## Competing interests

The authors declare no competing interests.
