## [Peer Review File · Nature Communications]

A wind-driven cold-energy harvesting device for cooling permafrost stratumREVIEWER COMMENTS

Reviewer #1 (Remarks to the Author):

To further enhance the manuscript, I suggest the following refinements:

1. Consider including a graphical representation illustrating the mechanism of the pump and wind turbine employed in the experiment. This visual aid will facilitate better comprehension and replication of the experimental setup by fellow researchers.
2. Provide detailed information regarding the local weather conditions during the experiments, including wind speed and air temperature. If the main text becomes overly cluttered, consider including this data in the supplementary materials section to ensure clarity in the manuscript.
3. It would be valuable to conduct a rough calculation of the heat flux of the double-tube device and compare it with existing thermosiphon systems. This comparison can be referenced from relevant literature, allowing readers to assess the effectiveness and efficiency of the proposed device in comparison to established cooling methods.
4. In Figure 2a, it would be beneficial to include centralizers in the outer annulus to ensure the inner tube remains centered within the larger tube. This would facilitate even flow of the coolant in the circumferential direction. Additionally, I suggest incorporating several horizontal circular sections in the figure to illustrate the internal structure of the fluid-circulation heat exchanger. These sections can provide a clearer depiction of the coolant backflow at the bottom of the fluid-circulation heat exchanger, highlighting its mechanism and aiding the reader's understanding.
5. It is intuitive to expect a time lag of 30-60 minutes between the temperature of the phase change material (PCM) in the clutch and the outer air temperature. It is important to address whether this lag has any influence on the operation of the device. Furthermore, it is worth noting that the temperature of the coolant held in the tube above the ground also exhibits a lag compared to the outer air temperature. Consequently, the inclusion of a clutch mechanism becomes highly desirable, as it ensures that coolant circulation initiates only after its temperature falls below the melting temperature of the PCM in the clutch. I recommend emphasizing this statement in the manuscript to highlight the significance of the clutch mechanism in the device's functionality.
6. In Figure 5, the manuscript does not provide information on the temperature below a depth of 8 meters. It would be more interesting and informative if the temperature at deeper levels could be included. I recommend that future studies focus on simulating the long-term temperatures of the ground where this device is installed, utilizing the temperature observations presented in this study as a reference. This will enable the verification of the model's accuracy. Additionally, it is important for the numerical model to predict the temperature distribution surrounding the tube, as this would provide valuable insights.
7. During the casting of concrete piles in permafrost regions, the permafrost soils surrounding the pile are susceptible to thawing due to the heat generated during the cement hydration process. This thawing significantly compromises the bearing capacity of the pile and cannot be naturally restored over time. To address this issue, I suggest an approach where, after the hydration process is complete, builders can circulate a low-temperature coolant (e.g., below 15 degrees) through the fluid-circulation heat exchanger. This coolant would flow from the inlet to the outer region, facilitating the rapid refreezing of the thawed soils within a short timeframe, ranging from days to weeks. Once the refreezing process is complete, the windmill and mechanical clutch can be connected to the fluid-circulation heat exchanger to preserve the permafrost foundation. Including this suggestion in the discussion section would not only enhance readers' understanding of the usefulness of the device but also promote the significance of this study within the academic field.

Reviewer #2 (Remarks to the Author):

Review of "A wind-driven cold-energy harvesting device for cooling permafrost stratum" by Qin et al.

The authors proposed a wind-driven cold-energy harvesting device that can be used to preserve permafrost. The novelty of this device is the combination of wind mill driven and fluid-circulation heat exchanger. The wind-driven system is similar to the pump system proposed by (Zhu et al.,

2023), and the fluid-circulation heat exchanger is similar to the traditional thermosyphons. Compared with the tilting or horizontal thermosyphons, this device may have better cooling capacity. However, it seems that its cooling capacity may be lower than that of vertical thermosyphons. That is because the device can activate when the outer air temperature exceeds the threshold of the phase change material and the weather should be windy, while the thermosyphons start to work only when a certain temperature difference between the air and ground temperature is reached. So, the operating time of this device may be less than the thermosyphons. Additionally, the more complex structure of this device than the thermosyphons may increase the failure rate, especially in the harsh environment of permafrost regions.

Specific comments:

Line 56–57: The crushed rock layers preserve underlying permafrost well in many cases, its disadvantage is that the cooling efficiency decreases if the void is filled by sand or weathered rock. The technique is cost-effective if the problem is solved.

Line 65-66: Why the combination of different technologies did not improve the cooling efficiency? Theoretically, some combination seems very promising.

Line 148-149: Why the phase change temperature is set to -4°C ? The operation time can be prolonged if the temperature is close to 0°C .

Line 161: The climatic condition of Beiluhe should be described, especially the air temperature and the wind speed, so the operation time of the device can be assessed.

Line 248-251: How about the annual operational duration of thermosyphons in a year? Many thermosyphons have been used on the Qinghai-Tibet Plateau, the annual operational duration of thermosyphons may be available. How do you calculate the annual operational duration? If the device only works 1 hour a day, does that count as an operational day?

Line 378-379: It may be inappropriate to define the thermosyphons as a natural air convection device. The condensation section (underground section) of thermosyphon is liquid and gaseous only exists in the evaporation section (overground end). The cooling of thermosyphons is because of the phase change of the refrigerant.

Line 419-421: The radius and temperature drop of the device should be compared with that of cooled thermosyphons.

Line 424-425: The four thermosyphons around a pile are installed not only to enhance the cooling efficiency but also to ensure the symmetry of ground temperature around the pile. Otherwise, the mechanical attributes of the surrounding soils may be uneven and result in the tilt of a pile. This aim may be also difficult to be achieved if only one set is used.

Zhu, Z., He, Z., Luo, F., Luo, B., Tang, C., Zou, Z., Guo, Z., Xiao, W., Jiang, X. and Li, L., 2023. Evaluating the performance of a novel ventilated embankment structure in warm permafrost regions by numerical simulation. *Cold Regions Science and Technology*, 209: 103805.

Responses to Reviewers' Comments

Dear editor and reviewers,

Thank you for considering our manuscript entitled “A wind-driven cold-energy harvesting device for cooling permafrost stratum” for potential publication in Nature Communications.

We are grateful for the insightful comments from the reviewers, which have greatly contributed to enhancing the quality of our manuscript. In our revised submission, we have carefully addressed each of the reviewers' remarks. For clarity, we've highlighted our responses in blue, with the original comments retained in black font.

Should you have any further inquiries or suggestions, please rest assured that we are committed to making the necessary revisions for the betterment of our manuscript.

Warm regards,

Yinghong Qin, Tianyu Wang, Weixin Yuan

REVIEWER COMMENTS

Reviewer #1 (Remarks to the Author):

To further enhance the manuscript, I suggest the following refinements:

1. Consider including a graphical representation illustrating the mechanism of the pump and wind turbine employed in the experiment. This visual aid will facilitate better comprehension and replication of the experimental setup by fellow researchers.

Response:

We extend our gratitude for your constructive feedback, which greatly aids in enhancing the quality of our manuscript.

In response, Fig. 1 has been refined to encapsulate a more detailed representation of the pump and wind turbine mechanisms. These modifications are discernible in Lines 113-115, and the revisions are visually depicted in the updated Fig. 2.

2. Provide detailed information regarding the local weather conditions during the experiments, including wind speed and air temperature. If the main text becomes overly cluttered, consider including this data in the supplementary materials section to ensure clarity in the manuscript.

Response:

We sincerely appreciate your insightful comment, which enhances the depth of our study. Such data is particularly pivotal when attempting simulations to gauge the

efficacy of the wind-driven cold-harvesting device.

In light of this, we have augmented the supplementary materials to incorporate local air temperature and wind speed. These additions encompass both graphical representations and raw data in an Excel file format. Kindly refer to the complementary file: S1 delineates the local air temperature, while S2 depicts the local wind speed.

3. It would be valuable to conduct a rough calculation of the heat flux of the double-tube device and compare it with existing thermosiphon systems. This comparison can be referenced from relevant literature, allowing readers to assess the effectiveness and efficiency of the proposed device in comparison to established cooling methods.

Response:

We extend our gratitude for your constructive feedback, emphasizing the significance of benchmarking the efficiency of our wind-driven cold-harvesting device against prevalent permafrost-cooling apparatuses, such as the thermosiphon.

In the amended manuscript, we quantified the power of the wind-driven heat-harvesting device and juxtaposed it with the empirical data presented by Zhang et al. (2022). This comparative analysis is discernible in Lines 398-405 of the main manuscript and is further elaborated in the S5 section of the supplementary materials.

4. In Figure 2a, it would be beneficial to include centralizers in the outer annulus to ensure the inner tube remains centered within the larger tube. This would facilitate even flow of the coolant in the circumferential direction. Additionally, I suggest incorporating several horizontal circular sections in the figure to illustrate the internal structure of the fluid-circulation heat exchanger. These sections can provide a clearer depiction of the coolant backflow at the bottom of the fluid-circulation heat exchanger, highlighting its mechanism and aiding the reader's understanding.

Response:

We express our appreciation for your constructive feedback, emphasizing the importance of enhancing the manuscript's visual clarity for readers.

In the updated version, we've introduced a horizontal circular section within the figure to distinctly illustrate the centralizer, as well as the downward and upward flow dynamics. Moreover, we've adjusted the figure's font for improved legibility and have overhauled its overall layout. We believe that these modifications will furnish readers with a more informative perspective. Should this reconfiguration not adequately convey our intentions or pertinent information, we stand prepared to undertake a comprehensive redesign of the figure.

Please refer to the modified Fig. 1 for these amendments.

5. It is intuitive to expect a time lag of 30-60 minutes between the temperature of the phase change material (PCM) in the clutch and the outer air temperature. It is important to address whether this lag has any influence on the operation of the device. Furthermore, it is worth noting that the temperature of the coolant held in the tube above the ground also exhibits a lag compared to the outer air temperature. Consequently, the inclusion of a clutch mechanism becomes highly desirable, as it ensures that coolant

circulation initiates only after its temperature falls below the melting temperature of the PCM in the clutch. I recommend emphasizing this statement in the manuscript to highlight the significance of the clutch mechanism in the device's functionality.

Response:

Your feedback is invaluable in enhancing the manuscript's rigor and coherence.

Indeed, there is a lag of approximately 15-30 minutes before the clutch disengages when the air temperature drops below the threshold. This delay is attributed to the thermal resilience of the tube wall and the intrinsic resistance of the Phase Change Material (PCM). Similarly, when the external air temperature transitions from a lower value surpassing the threshold, the inner coolant's temperature lags behind the external changes. Consequently, even if the external temperature rises beyond the threshold, the coolant's continuous circulation doesn't transfer excess warmth to the ground, as the inner coolant remains relatively cold. From a theoretical standpoint, it would be ideal for the PCM to occupy as minimal a volume as possible to ensure that its phase transition experiences minimal lag.

6. In Figure 5, the manuscript does not provide information on the temperature below a depth of 8 meters. It would be more interesting and informative if the temperature at deeper levels could be included. I recommend that future studies focus on simulating the long-term temperatures of the ground where this device is installed, utilizing the temperature observations presented in this study as a reference. This will enable the verification of the model's accuracy. Additionally, it is important for the numerical model to predict the temperature distribution surrounding the tube, as this would provide valuable insights.

Response:

Thank you for suggestion. It is true that in the current study, the temperature below the 8m is not measure so we do not know the temperature therein. In the future studies, it will be interesting to simulate the temperature below the lower tip of the device. It can be seen in Fig. 3 that only a small portion of the cold energy from the lower tip to the top tip is absorbed by the surrounding permafrost soils. It is thus expected that the wind-driven cold-energy harvesting device has a higher power if it is buried deeper. A simulation on the basis of the current measurement can be helpful to optimize the thermal parameters of the device and is helpful to improve the efficiency of the device. Such a simulation is also useful to understand the temperature evolution around the device and to visualize the cooling column around the device in a long run.

In the revision, the discussion section highlights the important of this simulation in future studies. Please see the revision in Line 495-498.

7. During the casting of concrete piles in permafrost regions, the permafrost soils surrounding the pile are susceptible to thawing due to the heat generated during the cement hydration process. This thawing significantly compromises the bearing capacity of the pile and cannot be naturally restored over time. To address this issue, I suggest an approach where, after the hydration process is complete, builders can circulate a low-temperature coolant (e.g., below 15 degrees) through the fluid-circulation heat

exchanger. This coolant would flow from the inlet to the outer region, facilitating the rapid refreezing of the thawed soils within a short timeframe, ranging from days to weeks. Once the refreezing process is complete, the windmill and mechanical clutch can be connected to the fluid-circulation heat exchanger to preserve the permafrost foundation. Including this suggestion in the discussion section would not only enhance readers' understanding of the usefulness of the device but also promote the significance of this study within the academic field.

Response:

We deeply value your suggestion, highlighting an area of the study that necessitates further exploration.

Indeed, our present study lacks temperature measurements below the 8m mark, rendering the conditions at those depths ambiguous. Future investigations would benefit from simulating temperatures beneath the device's lower tip. As illustrated in Fig. 3, a minuscule fraction of the cold energy between the lower and upper tips is absorbed by the adjacent permafrost soils. Thus, it's conceivable that embedding the wind-driven cold-energy harvesting device deeper could elevate its efficacy. Utilizing current measurements as a foundation for simulations could be instrumental in refining the device's thermal parameters and boosting its efficiency. Furthermore, such simulations would shed light on the temperature dynamics surrounding the device and offer insights into the long-term development of the cooling column.

The revised manuscript accentuates the significance of these potential simulations in the discussion section, visible in Lines 495-498.

Reviewer #2 (Remarks to the Author):

Review of “A wind-driven cold-energy harvesting device for cooling permafrost stratum” by Qin et al.

The authors proposed a wind-driven cold-energy harvesting device that can be used to preserve permafrost. The novelty of this device is the combination of wind mill driven and fluid-circulation heat exchanger. The wind-driven system is similar to the pump system proposed by (Zhu et al., 2023), and the fluid-circulation heat exchanger is similar to the traditional thermosyphons. Compared with the tilting or horizontal thermosyphons, this device may have better cooling capacity. However, it seems that its cooling capacity may be lower than that of vertical thermosyphons. That is because the device can activate when the outer air temperature exceeds the threshold of the phase change material and the weather should be windy, while the thermosyphons start to work only when a certain temperature difference between the air and ground temperature is reached. So, the operating time of this device may be less than the thermosyphons. Additionally, the more complex structure of this device than the thermosyphons may increase the failure rate, especially in the harsh environment of permafrost regions.

Zhu, Z., He, Z., Luo, F., Luo, B., Tang, C., Zou, Z., Guo, Z., Xiao, W., Jiang, X. and Li, L., 2023. Evaluating the performance of a novel ventilated embankment structure in

warm permafrost regions by numerical simulation. *Cold Regions Science and Technology*, 209: 103805.

Response:

Thank you for your discerning remarks, which have been instrumental in the enhancement of our manuscript.

In revisiting the work by Zhu et al., we took note of their innovative ventilation structure that employs forced convection to guide cold air into the ground. Although both investigations capitalize on wind energy for circulating cool in colder periods, marked distinctions become evident under detailed scrutiny. In our study, we deploy a liquid coolant as the medium for cooling, in contrast to Zhu et al.'s utilization of air. Furthermore, the circulatory systems in the respective studies differ: ours operates on a closed-loop basis, while Zhu et al. adopt an open system.

Addressing the matter of operating time, our calculations in Fig. 3i indicate an observed operational time of 0.2-0.3 years annually. Modifying the freezing temperature of the PCM in the clutch from -4°C to a range between -4°C and -1°C may augment the operating time. Yet, the implications of this modification on cooling efficiency warrant further exploration.

Turning our attention to power comparisons, our empirical study revealed that the wind-driven cold-energy harvesting device delivers an annual power of 68W. In contrast, a thermosyphon, in its active state, outputs 25W. Accounting for both active and dormant phases, the actual annual power of a thermosyphon would be marginally lower than 25W.

Concerning the device's intricate structure, we acknowledge that the wind-driven cold-energy harvesting device's architecture, encompassing a clutch, pump, and non-contract mechanical transfer system, is multifaceted. Future endeavors will delve into streamlining its components and diminishing the device's susceptibility to fatigue. The emphasis on such refinements is underscored in our discussion section, evident in Lines 427-439.

That said, it's pertinent to clarify that a thermosyphon's construction isn't necessarily straightforward. It encompasses an evaporative section, an adiabatic section, and a condenser section, replete with fins in the latter and specific textures on the adiabatic section. More significantly, operating akin to a pressure cooker, a thermosyphon's interior sustains high pressures necessitating a robust sealed system.

These nuanced considerations and comparisons have been embedded in our updated manuscript, as detailed in Lines 379-444.

Specific comments:

1. Line 56–57: The crushed rock layers preserve underlying permafrost well in many cases, its disadvantage is that the cooling efficiency decreases if the void is filled by sand or weathered rock. The technique is cost-effective if the problem is solved.

Response:

Thank you for pointing out this essential aspect.

Indeed, when crushed rock is supplanted with sand, there's an alteration in the thermal properties, primarily the cooling capacity of the rock layer, which gets

attenuated. Recognizing this, we have incorporated this important note into our manuscript to provide a more comprehensive understanding of the variables affecting the cooling capacity.

You can find this modification in the revised manuscript at Lines 43-45.

2. Line 65-66: Why the combination of different technologies did not improve the cooling efficiency? Theoretically, some combination seems very promising.

Response:

We appreciate your astute observation.

Indeed, while integrating various passive cooling techniques can offset the limitations intrinsic to individual methods, it doesn't inherently enhance the efficacy of permafrost cooling to sustainably maintain permafrost beneath expressways. We've updated our manuscript to encapsulate this notion more clearly.

This adjustment is reflected in Lines 67-70 of the revised manuscript.

3. Line 148-149: Why the phase change temperature is set to -4°C ? The operation time can be prolonged if the temperature is close to 0°C .

Response:

We greatly value your input, as it has been instrumental in refining our manuscript.

Our investigation into the wind-driven cold-energy harvesting device is at a nascent stage, and we're in the process of discerning its optimal cooling performance. The determination of the precise threshold temperature for maximizing cold energy harvesting for the underlying permafrost remains a challenge. If the threshold is set too low, it may limit the operational time and reduce the harvested cold energy. Conversely, setting it too high, for instance, at -1°C , might negate the benefits of colder energy deposits (like those below -5°C). Taking this potential counteraction into account, we initially set the threshold at -4°C .

Future research will involve experimenting with varying threshold temperatures to pinpoint the most effective temperature for maximum cold energy harvesting. This updated information has been incorporated into our manuscript, and can be consulted in Lines 256-261.

4. Line 161: The climatic condition of Beiluhe should be described, especially the air temperature and the wind speed, so the operation time of the device can be assessed.

Response:

We deeply appreciate your feedback, which has been crucial in enhancing the comprehensiveness of our manuscript.

In our revised version, we have incorporated a plot detailing the air temperature and wind speed at the Beiluhe permafrost testing site. This plot represents daily mean values and can be found in the supplementary document. Additionally, for those seeking a more granular perspective, we've provided air temperature and wind speed data recorded at minute intervals, accessible in the supplementary files.

To streamline the main manuscript, detailed information regarding local air temperature during the experiment can be referenced in the S1 section of the

complementary file, while data on local wind speed is available in the S2 section of the supplementary file.

5. Line 248-251: How about the annual operational duration of thermosyphons in a year? Many thermosyphons have been used on the Qinghai-Tibet Plateau, the annual operational duration of thermosyphons may be available. How do you calculate the annual operational duration? If the device only works 1 hour a day, does that count as an operational day?

Response:

Thank you for pointing out the potential ambiguity in our manuscript.

In the original description, we utilized Fig. 3i to showcase the device's operational hours. The device's operational status was determined by examining the temperature difference between the coolants in the inner and outer tubes. Specifically, when there was a discernible temperature differential between the two tubes (as exemplified in Fig. 3a), it indicated the device was in operation. Conversely, if there was no temperature differential, it signified the device was in a dormant state. To quantify this, if the device was active for an entire hour on a given day, this was recorded as 1/24 day. Similarly, a minute of operation would be denoted as 1/(24*60) day. A full 24-hour operation in a day means the device ran continuously without any interruption. As depicted in Fig. 3i, for the year 2021-2022, the device mostly operated less than 24 hours on most days, barring a few exceptions spanning 3-5 days. The aggregate operational time over the year, with a threshold temperature set at -4°C, was calculated as 0.228 years. Modifying the threshold temperature might indeed influence the operational time, but that hypothesis warrants further exploration.

We have updated and clarified this aspect in our manuscript in the revised Fig. 3i.

6. Line 378-379: It may be inappropriate to define the thermosyphons as a natural air convection device. The condensation section (underground section) of thermosyphon is liquid and gaseous only exists in the evaporation section (overground end). The cooling of thermosyphons is because of the phase change of the refrigerant.

Response:

Thank you for pointing out the need for clarification regarding thermosyphons in our manuscript.

Indeed, thermosyphons offer a more intricate and efficient mechanism compared to natural air convection devices. Their operation relies on the phase changes of the working fluid, such as ammonia, leveraging the latent heat associated with evaporative cooling and condensation. While this cooling mechanism involves convective heat transfer due to the movement of the vaporized working fluid, it's imperative to distinguish it from the typical "natural" air convection which operates without phase changes.

To provide clarity, we have revised our manuscript as per your suggestion. You can refer to the modifications in Lines 53-57: "In the thermosyphon system, heat from the surrounding soil is dissipated through the evaporative and condensing loop of ammonia. Due to the circulation of ammonia vapor, this process can still be classified

as air convection cooling, adhering to the principles of heat exchange.”

7. Line 419-421: The radius and temperature drop of the device should be compared with that of cooled thermosyphons.

Response:

We appreciate the feedback which offers avenues for refining our manuscript.

In our revised analysis, we juxtaposed the power metrics of the wind-driven cold-energy harvesting apparatus against the thermosyphon. While certain studies propose simulations of the thermosyphon's temperature decline and radius, we chose to foreground our analysis on experimental observations over simulations. To illustrate, Zhang et al. (2011) ¹ modelled an embankment heat flux at $100\text{W}/\text{m}^2$. Given an evaporative section with a 1.5m^2 ring area (spanning 6m in length and 0.08m in diameter), the projected annual mean power of the thermosyphon would approximate 150W. Yet, a subsequent study by Zhang et al. (2017) ² inferred an annual mean power of 60W based on a $30\text{W}/\text{m}^2$ heat flux, with an evaporative section extending 8m in length and retaining a diameter of 0.08m. These disparities might emanate from the inherent challenges of transposing three-dimensional phenomena onto two-dimensional simulations. Moreover, a recent experimental inquiry by Zhang et al. (2022) ³ discerned an instantaneous power of 25W for an activated thermosyphon.

Although we do not interrogate the variances among these findings, our intent remains to align experimental observations from diverse studies. We acknowledge the absence of comprehensive experimental insights on the thermosyphon's cooling radius but remain poised to incorporate any emerging data.

Readers can delve into a detailed comparative analysis between the thermosyphon and the wind-driven cold-energy harvesting device in our supplementary section (S5), and the pertinent synopsis on Lines 398-405 of the main document.

Reference:

1. Zhang M, Lai Y, Zhang J, Sun Z. Numerical study on cooling characteristics of two-phase closed thermosyphon embankment in permafrost regions. *Cold Reg Sci Technol* **65**, 203-210 (2011).
2. Zhang M, Pei W, Lai Y, Niu F, Li S. Numerical study of the thermal characteristics of a shallow tunnel section with a two-phase closed thermosyphon group in a permafrost region under climate warming. *Int J Heat Mass Transfer* **104**, 952-963 (2017).
3. Zhang M, Yan Z, Pei W, Lai Y, Qin Y, Yu F. Experimental study on the startup and heat transfer behaviors of a two-phase closed thermosyphon at subzero temperatures. *Int J Heat Mass Transfer* **190**, 122283 (2022).

8. Line 424-425: The four thermosyphons around a pile are installed not only to enhance the cooling efficiency but also to ensure the symmetry of ground temperature around the pile. Otherwise, the mechanical attributes of the surrounding soils may be uneven and result in the tilt of a pile. This aim may be also difficult to be achieved if only one set is used.

Response:

We appreciate the astute observation and feedback.

In deploying thermosyphons for cooling the footing of a power tower, they are strategically situated at the four corners to engender symmetrical cooling around the footing. This arrangement, however, inherently precludes the thermosyphons from being centrally placed — a position where heat dispersal would be truly symmetrical. This distinctive characteristic, wherein heat radiates symmetrically around the footing, accentuates the advantages of the wind-driven cold-harvesting device over traditional thermosyphons.

To elucidate this nuanced difference and avoid potential misinterpretation, we have made appropriate modifications in Lines 462-466 of the revised manuscript.

REVIEWERS' COMMENTS

Reviewer #1 (Remarks to the Author):

The manuscript has been modified and responded in a beneficial way. The experimental conditions and research results were supplemented. Proposed manuscript acceptance.

Reviewer #2 (Remarks to the Author):

I appreciate the authors for considering my comments to the previous manuscript. I believe that the manuscript is greatly improved as a result. I list my minor points below.

Line 124~125: Please explain in detail how the non-contact transmission works. The tight seal is crucial for the long-term operation of the device, the non-contact transmission may influence the efficiency of the fluid-circulation.

Line 153~155: The solar radiation heating can slow down the freezing of PCM in sunny weather, surface reflecting coating may be beneficial.

Line 214~215: The sentence seems confusing. To my understanding, the engage or disengage of the clutch depends on the freezing or thawing of PCM. The temperature difference between the inner and outer tube results from the fluid-circulation driven by the pump.

RESPONSES TO REVIEWERS' COMMENTS

Dear editor and reviewers,

I hope this letter finds you well.

Thank you for considering our manuscript entitled “A wind-driven cold-energy harvesting device for cooling permafrost stratum” for potential publication in *Nature Communications* (Submission ID: NCOMMS-23-23431A).

We are grateful for the insightful comments from the reviewers, which have greatly contributed to enhancing the quality of our manuscript. In our revised submission, we have carefully addressed each of the reviewers' remarks. For clarity, we've highlighted our responses **in blue**, with the original comments retained in black font.

Should you or the reviewers have any further inquiries or suggestions, please rest assured that we are committed to making the necessary revisions for the betterment of our manuscript.

Warm regards,

Yinghong Qin, Tianyu Wang, Weixin Yuan

Reviewer #1:

The manuscript has been modified and responded in a beneficial way. The experimental conditions and research results were supplemented. Proposed manuscript acceptance.

Response:

We would like to express our gratitude for your positive feedback and for recognizing the efforts we have made to address your previous comments. We are pleased to hear that the modifications and additions we made to the manuscript have been beneficial. Your acknowledgment of the supplemented experimental conditions and research results is encouraging. We are thrilled to receive your proposal for the acceptance of our manuscript. It is truly an honor to potentially contribute our work to *Nature Communications*. Once again, we appreciate your valuable feedback and recommendation for the acceptance of our manuscript.

Reviewer #2:

I appreciate the authors for considering my comments to the previous manuscript. I believe that the manuscript is greatly improved as a result. I list my minor points below.

Response:

Thank you for your constructive and insightful comments on our previous manuscript. We are grateful for the time and effort you have invested in reviewing our work, and we appreciate your positive feedback on the modifications we have made to the manuscript. We have carefully considered the minor points you have listed below, and we have made the necessary corrections and revisions. We hope that the changes we have made address your concerns and meet your expectations.

1. Line 124~125: Please explain in detail how the non-contact transmission works. The tight seal is crucial for the long-term operation of the device, the non-contact transmission may influence the efficiency of the fluid-circulation.

Response:

Thank you for your constructive comments. In this device, a number of neodymium-iron-boron magnets are fixed on the rotating shaft of the wind mill, on the clutch and on the rotating shaft of the pump, and are designed to rotate on concentric axes to avoid direct contact between components. Power transmission can be achieved under the action of the magnetic field, reducing the risk of seal damage caused by mechanical transmission. In the perennial permafrost regions of north-east China and the Qinghai-Tibetan Plateau, where wind speeds are high and temperatures are low in winter, wind mill can collect wind power and drive the clutch to rotate, thereby driving the pump. This contactless transmission has been explained in the revised manuscript, see lines 395-401 for details.

2. Line 153~155: The solar radiation heating can slow down the freezing of PCM in sunny weather, surface reflecting coating may be beneficial.

Response:

Thank you for your valuable comments. In certain weather conditions, especially on sunny days, excessive solar radiation can lead to an increase in ambient temperature, which can slow down the freezing process or even lead to partial melting of the PCM. To solve this problem, applying a reflective coating to the surface is indeed a viable solution, which helps to reduce the absorption of solar energy, minimising the amount of heat absorbed by both the PCM and the circulating coolant. This in turn improves the freezing efficiency and maintains the performance of the equipment in sunny weather conditions.

Indeed, in our experiments we coated the surface of the appliance with a low-reflective coating (see below), which significantly improved the efficiency of the appliance. Fig. 2b in the manuscript also confirms our practice. Unfortunately, due to an oversight, we did not mention this practice in the previous manuscript. This has now been added in lines 446-448 of the revised manuscript.

Fig.R1 Whitewash surfaces to minimize the effects of solar radiation on the clutch.

3. Line 214~215: The sentence seems confusing. To my understanding, the engage or disengage of the clutch depends on the freezing or thawing of PCM. The temperature difference between the inner and outer tube results from the fluid-circulation driven by the pump.

Response:

Thank you for pointing out the confusion in lines 214-215 of our manuscript. We apologize for the unclear sentence and thank you for understanding our original intent.

Your understanding is correct. The engagement or disengagement of the clutch does depend on the freezing or thawing of the PCM. The PCM freezes and then solidifies, allowing the mechanical clutch to engage and transfer torque from the wind turbine to the fluid circulation heat exchanger. This engagement allows the pump to circulate fluid which creates a temperature difference between the inner and outer tubes.

We agree with your clarification that the temperature difference between the inner and outer tubes is caused by the pump-driven circulation of fluid. When the PCM thaws, the device disengages and the fluid circulation ceases; in this case, the temperature difference between the inner and outer tubes vanishes.

What we are trying to convey is **"If there is a significant temperature difference between the coolant in the inner tube and that in the outer annulus, it indicates that the pump is effectively circulating the coolant. This means that the cold energy is being transferred to the soil through the outer tube, indicating that the clutch is engaged. However, if the temperature difference is minimal, the clutch remains disengaged."**

To avoid further confusion, we have revised this sentence to clearly express the relationship between clutch engagement/disengagement and PCM freezing/thawing, and to emphasize the role of pump-driven fluid circulation in establishing the temperature difference between the inner and outer tubes. See lines 115-119 in the latest version.

Thank you for bringing this to our attention, and we appreciate your feedback to improve the accuracy and clarity of our manuscript.